# Prevalence of Function-Dependent Temporomandibular Joint and Masticatory Muscle Pain, and Predictors of Temporomandibular Disorders among Patients with Lyme Disease

**DOI:** 10.3390/jcm8070929

**Published:** 2019-06-28

**Authors:** Magdalena Osiewicz, Daniele Manfredini, Grażyna Biesiada, Jacek Czepiel, Aleksander Garlicki, Ghizlane Aarab, Jolanta Pytko-Polończyk, Frank Lobbezoo

**Affiliations:** 1Department of Integrated Dentistry, Dental Institute, Faculty of Medicine, Jagiellonian University Medical College, 31-155 Krakow, Poland; 2School of Dentistry, University of Siena, 53100 Siena, Italy; 3Department of Infectious and Tropical Diseases, Faculty of Medicine, Jagiellonian University Medical College, 31-155 Krakow, Poland; 4Department of Orofacial Pain and Dysfunction, Academic Centre forDentistry Amsterdam (ACTA), University of Amsterdam and Vrije Universiteit Amsterdam, 1081 LA Amsterdam, The Netherlands

**Keywords:** palpation tests, temporomandibular disorders, static/dynamic tests, comorbidity, Lyme disease

## Abstract

The aim was to determine the occurrence of temporomandibular disorders (TMDs) in patients with Lyme disease (LD), and to estimate the contribution of factors that may identify TMD among LD patients. In seventy-six (*N* = 76) adult patients with LD (mean age 57.6 ± 14.6 years) and 54 healthy non-Lyme volunteers with a mean age of 56.4 ± 13.5 years, possible function (i.e., non-pain) diagnoses were established using the Research Diagnostic Criteria of Temporomandibular Disorders (RDC/TMD). Pain diagnoses were established by means of the function-dependent dynamic and static tests. The two groups did not significantly differ in the frequency of disc displacements diagnoses and function-dependent pain diagnoses. LD showed a significantly higher frequency (*p* < 0.001) of osteoarthrosis than the control group. For the prediction of pain diagnoses in LD patients, the single regression analyses pointed out an association with age, sleep bruxism (SB), and awake bruxism (AB). Two predictors (i.e., SB (*p* = 0.002) and AB (*p* = 0.017)) were statistically significant in the final multiple variable model. The frequency of TMD in patients with LD based on function-dependent tests was not significantly different from that in the control group. This investigation suggests that the contribution of bruxism to the differentiation between patients with Lyme and TMD is high.

## 1. Introduction

Lyme disease (LD) is the most common tick-borne disease caused by bacteria spirochetes of the *Borrelia* species, classified as *Borrelia burgdorferi* (Bb) strain [1,2]. Most cases of LD are reported in the northern hemisphere, mainly in the US and Europe [3,4,5,6]. Vector ticks are spirochetes of the genus *Ixodes.* An average LD incubation period takes from 3 to 30 days, although it might be prolonged up to 3 months. The typical clinical picture of LD may include skin, joints, heart, and/or nervous system symptoms, less often symptoms involving the eyes [7].

Depending on the symptoms and their time of onset, LD is classified in Lyme neuroborreliosis (LN) or Lyme arthritis (LA), and in early localized, early disseminated, or late disseminated infection. Symptoms may affect more than one organ, also including skin and eyes [8,9]. Lyme neuroborreliosis occurs most commonly as root pain, lymphocytic meningitis, cranial nerve palsies, especially involving nerve VII, or less frequently as peripheral nerve palsies [8,9]. LA usually affects large joints (e.g., the knees) and is usually asymmetrical. Arthritis can occur in other joints, including the shoulders, elbows, wrists, and ankles [9,10] and manifests with a slight swelling and pain [7,8,9].

The most common non-dental orofacial pain conditions are temporomandibular disorders (TMDs) [11]. Apart from pain, TMD signs include limitation of mouth opening and joint sounds. Similar symptoms can be caused also by other orofacial or systemic disorders, even including LD. There are limited studies regarding TMD involvement in patients with LD, hence the relationship between LD and TMD is still unclear. A temporomandibular joint involvement was frequently reported in early studies of LA [12,13], however it has been mentioned less often over the current years. In subsequent studies of Heir and Fein, Wolańska-Klimkiewicz et al., and te Veldhuis et al., an increase in reporting jaw-muscle symptoms in LD patients was noticed [14,15,16].

Based on previous research by Koutris et al. and Osiewicz et al., a differential diagnosis between TMD and comorbid diseases is implemented by the use of function-dependent tests. In particular, using static/dynamic tests may help reducing the false positives [17,18,19] as well as building a prediction model to assess which factors predict the presence of TMD symptoms in combination with comorbid disease, such as LD. This would be of help for the dentists to refer a patient to a proper specialist for a diagnosis and LD treatment. It would also help limiting delays in interventions and/or unnecessary treatment.

Considering these drawbacks of previous studies and the scarceness of research on this topic, the aims of the present study were: To assess the occurrence of TMD symptoms in patients with LD;To compare the occurrence of TMD symptoms in patients with LN and LA;To estimate the contribution of various factors that may predict TMDs in LD patients.

## 2. Materials and Methods

### 2.1. Participants

The study was conducted at the Department of Infectious and Tropical Diseases of the University Hospital in Krakow, Poland, on 76 consecutive adult patients with LD seen between the years 2013 and 2016 (mean age 57.6 ± 14.6) and 54 healthy non-LD volunteers with a mean age of 56.4 ± 13.5 years. Volunteers of the control group were consecutively recruited from among the patients of the Dental Institute of the Jagiellonian University in Krakow, Poland. All subjects were aged over 18 years and gave informed consent.

Based on a medical history and clinical examination, the presence of LD was excluded from the control group. The infection was confirmed by examining the presence of specific antibodies in serum and/or cerebrospinal fluid (CSF) using ELISA test (Biomedica, Vienna, Austria), and the borderline or positive results in serum were confirmed with the use of Western-blot test (recomBlot Borrelia, Microgen, London, UK). In the group of patients with suspected LN, lumbar puncture was performed, examining the level of cytosis, protein concentration, glucose, and chlorides [8,9]. The patients were divided into two groups: LA and LN. Diagnostic tests for LD were carried out by an infectious disease specialist (GB).

The study was conducted according to the principles of the Helsinki Declaration and approved by the Bioethics Committee of the Jagiellonian University (No. KBET/200/B/2011).

### 2.2. Study Design

During a dedicated session at the Department of Infectious and Tropical Diseases, a specialist in TMD (MO) performed an examination according to the validated Polish version of Research Diagnostic Criteria for Temporomandibular Disorders (RDC/TMD) guidelines [20,21,22]. All participants also completed the RDC/TMD history-taking questionnaire and the Axis II to assess potential predictors: Sleep bruxism (SB), awake bruxism (AB), somatisation (SOM), depression (DEP), graded chronic pain scale (GCPS), other painful joints (OPJ), tinnitus, headaches or migraines (HM), education (high education or no high education), gender, and age. The examiner was trained in performing RDC/TMD clinical examination and dynamic/static tests by a calibrated examiner (FL) during a 3-year TMD and Orofacial Pain specialization program at the Department of Orofacial Pain and Dysfunction, Academic Centre for Dentistry Amsterdam (ACTA) [23]. Each patient had a panoramic radiograph provided by the Dental Institute of the Jagiellonian University in Krakow to exclude pain of possible dental origin.

In the next phase, the RDC/TMD was used to establish possible function (i.e., non-pain) diagnoses of the masticatory system. IIA disc displacement with reduction: The disc is displaced from its position between the condyle and the fossa to an anterior and medial or lateral position, but reduces during full opening; IIB disc displacement without reduction with limited mouth opening: A condition in which the disc is displaced from its normal position between the condyle and the fossa to an anterior and medial or lateral position, associated with limited mandibular opening; IIC disc displacement without reduction, without limited opening: A condition in which the disc is displaced from its normal position between the condyle and the fossa to an anterior and medial or lateral position, not associated with limited opening; and IIIC osteoarthrosis: Coarse crepitus in the temporomandibular joint (TMJ) and/or tomograms showing pathology in the TMJ [20,21,22].

For pain diagnoses, function-dependent dynamic and static tests were performed, following the methods that were previously described by Visscher et al. and Osiewicz et al. [18,19,24]. Those tests aim to elicit function-dependent pain in the temporomandibular joints and/or masticatory muscles through loaded movements of the mandible and heavy static muscular effort, respectively, and are an expansion of the RDC/TMD protocol. Based upon the measurements derived from the clinical examination, the following TMD-pain diagnoses could be established: Mainly myogenous pain (MMP) and mainly arthogenous pain (MAP).

### 2.3. Data Analysis

Categorical data were expressed as number and percentage, and comparisons between LD and controls and between LA and LN were performed using chi-square (χ^2^) or Fisher exact tests. The single logistic regression model was estimated for the different independent variables. Only those factors that were significant at *p* < 0.1 in the single-variable logistic regression analysis were included in the initial multiple variable regression model. Then, the variable with the weakest association was removed from the model. This procedure was repeated until all the variables that were retained in the model showed a *p* < 0.05. The odds ratios (OR) for MMP and/or MAP were assessed for each variable. OR values higher than 2 are commonly considered significant from a clinical viewpoint. Nagelkerke’s R-square was obtained as an estimation of the total variance explained by the predictor factors included in the model.

Statistics were performed using StatsDirect, version 2.8.0 (StatsDirect Ltd., Cambridge, UK).

## 3. Results

LD patients and control group did not differ significantly in the frequency of IIA diagnosis. Any subjects in the control group or in the LD patients group showed signs of IIB and IIC diagnoses. LD patients showed a significantly higher frequency (*p* < 0.001) of IIIC diagnosis than the control group. The frequency of MMP and/or MAP in LD patients (18.4%) was not significantly different in comparison to the control group (7.4%). The result was significant at the 10% level, but not at the 5% level (*p* = 0.059).

The frequency of SB and AB in the two groups was not significantly different. In the LD group, moderate or severe SOM was shown in 73.7% of patients, whilst in the control group the frequency was 7.4% (*p* < 0.001). The frequency of high levels of chronic pain-related impairment (i.e., GCPS grades III or IV) was also significantly different between the LD patients and the control group at the level of *p* = 0.002. Chronic pain-related impairment in the LD group was shown in 15.8% of patients, whilst none of the control participants showed this. There were significantly more patients with moderate and severe DEP (55.3%) in the LD group than in the control group (7.4%; *p* < 0.001). Similar differences were observed in the case of OPJ (LD group: 46%, control group: 14.8%), tinnitus (53.9% in the LD group and 3.7% in the control group), and HM (64.5% in the LD group and 11.1% in the control group) (Table 1).

In the LD group, half of the patients (*N* = 38) were diagnosed with LA and half with LN. The two groups did not significantly differ in the frequency of RDC/TMD-based function (non-pain) diagnoses, nor in that of the function-dependent pain diagnoses. The frequency of self-reported AB was significantly higher in the group of patients with LA (34.2%) than with LN (10.5%) (*p* = 0.013). Similar differences were observed for moderate and severe SOM, which was more frequent in patients with LA (84.21%) than with LN (63.16%) (*p* = 0.033). The frequency of moderate and severe DEP was also significantly higher in the LA group than in the LN group (65.8% vs. 44.7%). Similar differences (*p* = 0.010) were observed for OPJ (LA, 60.5%; LN, 31.6%) (Table 2).

For the prediction of MMP and/or MAP in LD, the single regression analyses showed a possible association with age, SB, and AB. Two predictors (i.e., SB (*p* = 0.002) and AB (*p* = 0.017)) were statistically significant in the final multiple variable model that best predict the presence of positive outcomes of functional pain tests, viz., MMP and/or MAP in the LD group. Both SB and AB significantly increased the risk for MMP and/or MAP, with an odds ratio of 6.4 for AB and for 18.8 for SB (Table 3).

## 4. Discussion

Findings of earlier studies suggest that dynamic/static tests should be used as a part of the routine TMD assessment [17,18,19]. An investigation by Osiewicz et al. in the population of patients with LD showed that 70% of them tested positive for RDC/TMD diagnosis of myofascial pain. Based on that, it might be suggested that palpation tests may be positive in TMD disorders of muscular origin in patients with other know conditions, such as LD, possibly due to the fact that the presence of comorbid conditions causes widespread muscle sensitization. Within these premises, the first aim of this study was to assess the occurrence of TMDs in patients with LD and to assess possible differences between patients with LA and LN. Function (i.e., non-pain) diagnoses of the masticatory system were based on RDC/TMD, whist pain diagnoses were based on the function-dependent dynamic and static tests. The second aim was to estimate the contribution of various factors that might identify TMD in LD patients.

The frequency of TMD was not significant in patients with LD in comparison with the control group as well as between LA and LN. It was only in the case of IIIC RDC/TMD diagnosis that Lyme patients showed a significantly higher frequency. A broadly accepted definition of LA states that an inflammation applies mostly to large joints [25], but a TMJ might be also inflamed. LD usually results in arthritis, but long lasting and untreated LA can cause a destruction of the joint, also known as osteoarthrosis.

This investigation suggests that concurrent bruxism report may be the main factor to predict the presence of function-dependent TMD pain in patients with LD, to discriminate from those with Lyme-related pain. The risk for positive static/dynamic and/or palpation test in Lyme patients is much higher when the patients report sleep and/or awake bruxism. The role of bruxism in the etiology of TMDs is a much debated topic, because of the controversial relationship between self-reported bruxism and TMD symptoms as well as the difficulties to identify a dose–response relationship. Thus, it has become one of the most controversial issues within dental literature [26,27].

The absence of data on function-dependent TMD symptoms in LD makes it impossible to compare the present findings with the available literature. Earlier studies on TMD occurrence in patients with other primary conditions, such as irritable bowel syndrome (IBS) or Hashimoto thyroiditis, which usually showed a higher frequency of TMDs than in the general population, were based on palpation tests only [28,29]. Thus, such findings are potentially related to generalized hypersensitivity and not only to local musculoskeletal complaints [17,18,19,30].

There are several factors that might have affected the outcomes of this investigation. The main problem in this study was that the regression model included factors that were not assessed based on the best available methods. For instance, awake and sleep bruxism were just screened with the questions included in the history-taking questionnaire of the RDC/TMD. Future studies should assess AB by means of an Ecological Momentary Assessment, and SB by means of at least electromyography [31,32,33]. Another critical issue is that dynamic/static tests are not included in the standard protocols to diagnose TMD pain yet; they are only recommended as possible complimentary tools [34]. Thus, a suitable training in the specific techniques and in the verbal instructions for the patients is necessary. The adoption of the updated DC/TMD version plus the complimentary function-dependent tests is a recommended strategy to further increase the strength of similar data for cross-cultural comparison.

As an additional finding, it is remarkable that the psychosocial status of patients with LD was compromised in comparison to the control group. However, function-dependent tests are not potentially related to a generalized hypersensitivity, so it can be hypothesized that such psychosocial status was related with the presence of LD itself.

On the other hand, one of the strengths of this study is that for the first time, it assessed the frequency of TMD symptoms based on function-dependent static/dynamic tests. It also suggested that, even in the presence of concurrent conditions, the main factor that increases the risk for function-dependent TMD-pain diagnosis has to be searched in the complex spectrum of bruxism activities. The clinical significance of the study’s results confirms that dynamic/static tests should be used as a part of the routine TMD assessment. If the tests are negative, the patient should be referred to an infectious disease specialist to save valuable time spent on unnecessary treatment and to prevent delay of the appropriate intervention.

## 5. Conclusions

With respect to the threefold aim, the results of this study suggest that the frequency of TMD was not significant in the patients with LD and that there are no differences between patients with LA or LN. This investigation suggests that the impact of bruxism to predict the presence of TMD pain in patients with LD is high.

## Figures and Tables

**Table 1 jcm-08-00929-t001:** Frequency of the Research Diagnostic Criteria of Temporomandibular Disorders (RDC/TMD)-based diagnosed of temporomandibular joint (TMJ) status, function-dependent tests, and of Axis II findings including sleep bruxism (SB) and awake bruxism (AB) in Lyme patients and control group.

	Lyme Patients		Control		*p* = Value
**IIA**	8	10.5%	7	12.96%	*p* = 1
**IIB/IIC**	0	0,0%	0	0.0%	*p* = 1
**IIIC**	13	17.1%	3	5.6%	*p* < 0.001
**MMP and/or MAP**	14	18.4%	4	7.4%	*p* = 0.059 *
**SB**	16	21%	7	12%	*p* = 0.234
**AB**	17	22.4%	9	16.7%	*p* = 0.423
**SOM**	56	73.7%	4	7.4%	*p* < 0.001
**GCPS**	12	15.8%	0	0.0%	*p* = 0.002
**DEP**	42	55.3%	4	7.4%	*p* < 0.001
**OPJ**	35	46%	8	14.8%	*p* < 0.001
**tinnitus**	41	53.9%	2	3.7%	*p* < 0.001
**HM**	49	64.5%	6	11.1%	*p* < 0.001

Disc displacement with reduction (IIA); disc displacement without reduction with limited mouth opening (IIB); disc displacement without reduction, without limited opening (IIC); osteoarthrosis (IIIC); mainly myogenous pain (MMP) and/or mainly arthogenous pain (MAP); sleep bruxism (SB); awake bruxism (AB); moderate/severe somatisation (SOM); high levels of chronic pain-related impairment (GCPS); moderate/severe depression (DEP); other painful joints (OPJ); tinnitus; headaches or migraines (HM).* Fisher exact test.

**Table 2 jcm-08-00929-t002:** Frequency of the RDC/TMD-based diagnosed of TMJ status, function-dependent tests and Axis II findings including SB and AB in Lyme arthritis (LA) and Lyme neuroborreliosis (LN) patients.

	LA (*n* = 38)		LN (*n* = 38)		*p*-Value
**IIA**	4	10.5%	4	10.5%	*p* = 1
**IIB**	0	0.0%	0	0,0%	*p* = 1
**IIC**	0	0.0%	0	0,0%	*p* = 1
**IIIC**	9	23.7%	4	10.5%	*p* = 0.128
**MMP and/or MAP**	8	21%	6	15.8%	*p* = 0.554
**SB**	10	26.3%	6	15.8%	*p* = 0.260
**AB**	13	34.2%	4	10.5%	*p* = 0.013
**SOM**	32	84.2%	24	63.2%	*p* = 0.033
**GCPS**	6	15.8%	6	15.8%	*p* = 1
**DEP**	25	65.8%	17	44.7%	*p* = 0.050
**OPJ**	23	60.5%	12	31.6%	*p* = 0.010
**tinnitus**	21	55.3%	20	52.6%	*p* = 0.818
**HM**	27	71%	22	57.9%	*p* = 0.231

**Table 3 jcm-08-00929-t003:** Single and multiple logistic regression models for the prediction of mainly myogenous pain (MMP) and/or mainly arthogenous pain (MAP) in Lyme disease (LD) patients. For each single regression, the number of cases included in the analysis is shown. Associations are expressed as odds ratio (OR) and 95% confidence interval (CI). For each removed predictor variable, the *p*-to-Exit is reported.

Predictor Variable	Number	Single Regression		Multiple Regression (*N* = 76)
*p*-Value	OR	95% CI	*p* to Exit	*p*-Value	OR	95% CI
Gender	female	35	*p* = 0.791	1.2	0.4	3.8	*p* = 0.790				
male	41									
Age		76	*p* = 0.008	1	0.9	1	*p* = 0.023	*p* = 0.173	1	0.9	1.0
Education	no	55	*p* = 0.166	2.4	0.7	7.9	*p* = 0.172				
yes	21									
SB	no	60	*p* < 0.001	23.3	5.6	97.8	*p* < 0.001	*p* ≤ 0.001	18.8	3.7	96
yes	16									
AB	no	59	*p* < 0.001	12.2	3.2	45.6	*p* < 0.001	*p* = 0.027	6.4	1.2	33
yes	17									
SOM	no	20	*p* = 0.647	1.4	0.3	5.6	*p* = 0.064				
yes	56									
GCPS	no	64	*p* = 0.525	1.6	0.4	6.9	*p* = 0.535				
yes	12									
DEP	no	34	*p* = 0.455	1.6	0.5	5.3	*p* = 0.449				
yes	42									
OPJ	no	41	*p* = 0.743	1.2	0.4	3.9	*p* = 0.743				
yes	35									
Tinnitus	no	35	*p* = 0.1371	0.4	0.1	1.3	*p* = 0.129				
yes	41									
HM	no	27	*p* = 0.2313	2.3	0.6	9.2	*p* = 0.208				
yes	49

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
