# Peer review of "Prevalence of Function-Dependent Temporomandibular Joint and Masticatory Muscle Pain, and Predictors of Temporomandibular Disorders among Patients with Lyme Disease"

_jcm, 2019, doi:10.3390/jcm8070929_

Round 1
Reviewer 1 Report
work is very interesting.
it seems to be strongly related to RDC/TMD guidelines used to assess and to perform the research. It might be advisable to report the RDC/TMD guidelines (Polish version) you used because in references only Poland version was reported. A table with assets could be recommended in order to better understand methodological choices. Please note that RDC/TMD were updated in latter versions.
Study design section
"During a dedicated session at the Department of Infectious and Tropical Diseases, a specialist
in TMD (MO) performed an examination according to the validated Polish version of Research
Diagnostic Criteria for Temporomandibular Disorders (RDC/TMD) guidelines [20-22]". Please clarify how examination was performed.
Author Response
Dear Reviewer
Thank you for all your comments, which we found very helpful and constructive. We have complied with the Reviewers’ requests and made all the necessary changes and corrections in the manuscript.
We hope it will meet your expectations, and look forward to your decision.
1. Concern of the reviewer:
it seems to be strongly related to RDC/TMD guidelines used to assess and to perform the research. It might be advisable to report the RDC/TMD guidelines (Polish version) you used because in references only Poland version was reported. A table with assets could be recommended in order to better understand methodological choices. Please note that RDC/TMD were updated in latter versions.
Study design section
"During a dedicated session at the Department of Infectious and Tropical Diseases, a specialistin TMD (MO) performed an examination according to the validated Polish version of ResearchDiagnostic Criteria for Temporomandibular Disorders (RDC/TMD) guidelines [20-22]". Please clarify how examination was performed.
Our response: This is a very good point. We would like to clarify that we do refer to Dworkin & LeResche, hence to the original version; not only to the Polish one. Based on your request, we added in the manuscript some sentences about the RDC/TMD diagnoses; the examination itself is described extensively in the references 20-22. Since we have added extra information to the text, which makes a table redundant. We are aware that RDC/TMD were updated in latter versions, but the Polish translation was not available at the time of the conception of this study.
Revised text: In the next phase, the RDC/TMD was used to establish possible function (i.e., non-pain) diagnoses of the masticatory system: IIA disc displacement with reduction: the disc is displaced from its position between the condyle and the fossa to an anterior and medial or lateral position, but reduces during full opening; IIB disc displacement without reduction with limited mouth opening: a condition in which the disc is displaced from its normal position between the condyle and the fossa to an anterior and medial or lateral position, associated with limited mandibular opening; IIC disc displacement without reduction, without limited opening: a condition in which the disc is displaced from its normal position between the condyle and the fossa to an anterior and medial or lateral position, not associated with limited opening; and IIIC osteoarthrosis: coarse crepitus in the TMJ and/ or tomograms showing pathology in the TMJ [20-22]. For a detailed description of the examination methods, see [20-22].
Reviewer 2 Report
In this study, the authors aimed to: (1) assess the occurrence of symptoms of Temporomandibular disorders (TMD) in patients with Lyme disease (LD), (2) compare the occurrence of TMD symptoms in patients with Lyme-induced arthritis (LA) or Lyme neuroborreliosis (LN), and (3) estimate the contribution of various factors that may predict TMDs in LD patients. Statistical analyses revealed no significant differences in the frequencies of TMD symptoms between TMD patients with and without LD, and no differences in those who were diagnosed with LA and LN. They found a higher incidence of bruxism in TMD patients with LD, implicating possible attribution of sleep or awake bruxism in TMD.
Lyme disease is the most common tick-borne disease and results from infection of bacterial spirochetes. The clinical picture of patients diagnosed with LD includes headache, chronic migraines, neck and shoulder pain and long-term complications affecting synovial joints including TMJs. Interestingly, some of these symptoms are also common to patients afflicted with TMD. Therefore, it is important and critical for dentists and medical doctors to distinguish these two diseases in the clinical setting. Unfortunately, this study did not provide strong clinical evidence which will allow clinicians to definitively distinguish between LD and TMD in their patients. However, given that this study reminds this reviewer that Lyme’s disease continues to be under diagnosed and possibly missed in the clinic, this report may encourage Clinicians to reevaluate the diagnostic importance of these two diseases.
Specific comments
1. It might be useful to include patient history for TMD and whether or not there are any changes in TMD symptoms after LD infection.
2. The authors show that TMJ osteoarthritis is significantly higher in LD patients. What is the potential role of LD in this condition?
Author Response
Dear Reviewer
Thank you for all your comments, which we found very helpful and constructive. We have complied with the Reviewers’ requests and made all the necessary changes and corrections in the manuscript.
We hope it will meet your expectations, and look forward to your decision.
1. Concern of the reviewer: It might be useful to include patient history for TMD and whether or not there are any changes in TMD symptoms after LD infection.
Our response: This is surely a very nice observation and valid suggestion for future studies. Unfortunately, we did not collect patients’ history of TMD.
Revised text: not applicable
2. Concern of the reviewer: The authors show that TMJ osteoarthritis is significantly higher in LD patients. What is the potential role of LD in this condition?
Our response: Thank you for your query. We added a sentence to the discussion concerning the possible association between osteoarthrosis and LD.
Revised text: A broadly accepted definition of LA states that an inflammation applies mostly to large joints (Arvikar and Steere et al 2015), but a TMJ might be also inflamed. LD usually results in arthritis, but long lasting and untreated LA can cause a destruction of the joint, also known as osteoarthrosis.
Arvikar S., Steere A.: Diagnosis and Treatment of Lyme Arthritis. Infect Dis Clin N Am 29 (2015) 269-280.
Reviewer 3 Report
This study deals with the occurrence of non-dental orofacial pain condition TMDs, in patients with Lyme disease. They also analyze the contribution factors to identify TMD among LD patients. The study is original, interesting and increase the present knowledge in this area.
Comments
1. Please provide full for SB and AB in the abstract.
2. Better to discuss the clinical significance of the findings in the discussion.
Author Response
Dear Reviewer
Thank you for all your comments, which we found very helpful and constructive. We have complied with the Reviewers’ requests and made all the necessary changes and corrections in the manuscript.
We hope it will meet your expectations, and look forward to your decision.
1. Concern of the reviewer: Please provide full for SB and AB in the abstract.
Our response: Thank you for your suggestion. We added sleep bruxism and awake bruxism in the abstract.
Revised text: For the prediction of pain diagnoses in LD patients, the single regression analyses pointed out an association with age, sleep bruxism (SB), and awake bruxism (AB).
1. Concern of the reviewer: Better to discuss the clinical significance of the findings in the discussion.
Our response: Thank you for your request. Based on that, we added a sentence to the discussion concerning the clinical significance.
Revised text: The clinical significance of the study’s results confirms that dynamic/static tests should be used as a part of the routine TMD assessment. If the tests are negative, the patient should be referral to a infectious disease specialist, to save valuable time spent on unnecessary treatment and to prevent delay of the appropriate intervention.